

# Novel approach for quantitative and qualitative authors research profiling using feature fusion and tree-based learning approach

Muhammad Umer[1], Turki Aljrees[2], Saleem Ullah[1] and Ali Kashif Bashir[3]

[1] Department of Computer Science, Khwaja Fareed University of Engineering & IT, Rahim Yar Khan, Punjab, Pakistan
[2] Department of Computer Science and Engineering, University of Hafr Al-Batin, Hafar Al-Batin, Saudi Arabia
[3] Department of Computing and Mathematics, The Manchester Metropolitan University, Manchester, United Kingdom

## ABSTRACT

Article citation creates a link between the cited and citing articles and is used as a basis for several parameters like author and journal impact factor, H-index, i10 index, *etc.*, for scientific achievements. Citations also include self-citation which refers to article citation by the author himself. Self-citation is important to evaluate an author's research profile and has gained popularity recently. Although different criteria are found in the literature regarding appropriate self-citation, self-citation does have a huge impact on a researcher's scientific profile. This study carries out two cases in this regard. In case 1, the qualitative aspect of the author's profile is analyzed using hand-crafted feature engineering techniques. The sentiments conveyed through citations are integral in assessing research quality, as they can signify appreciation, critique, or serve as a foundation for further research. Analyzing sentiments within in-text citations remains a formidable challenge, even with the utilization of automated sentiment annotations. For this purpose, this study employs machine learning models using term frequency (TF) and term frequency-inverse document frequency (TF-IDF). Random forest using TF with Synthetic Minority Oversampling Technique (SMOTE) achieved a 0.9727 score of accuracy. Case 2 deals with quantitative analysis and investigates direct and indirect self-citation. In this study, the top 2% of researchers in 2020 is considered as a baseline. For this purpose, the data of the top 25 Pakistani researchers are manually retrieved from this dataset, in addition to the citation information from the Web of Science (WoS). The self-citation is estimated using the proposed model and results are compared with those obtained from WoS. Experimental results show a substantial difference between the two, as the ratio of self-citation from the proposed approach is higher than WoS. It is observed that the citations from the WoS for authors are overstated. For a comprehensive evaluation of the researcher's profile, both direct and indirect self-citation must be included.

Corresponding author
Saleem Ullah,
saleem.ullah@kfueit.edu.pk

## INTRODUCTION

Scientific publications have increased substantially over the past decade and a large number of researchers are publishing their work in the form of articles and books worldwide. Consequently, a large number of publications are available today with varying scientific quality and impact. Therefore, the necessity of reviewing and rating published scientific articles is in great demand. A multitude of criteria for evaluating the quality of a scientific article may be found in the literature. One of the most important assessment metrics is citation count which is an important parameter since it is frequently used to assess a paper's or researcher's impact (*Garfield, 1998*; *Herther, 2009*; *Oppenheim, 1997*). In addition, it has served as the foundation for other additional measures, including the h-index (*Hirsch, 2005*), i-10 index, impact factor (*Garfield (2006)* and other assessment metrics for researchers, conferences, journals, and research institutions (*Moed et al., 2012*; *Wildgaard, Schneider & Larsen, 2014*).

The bibliometric measurements are the most significant application of citation sentiment analysis. The analysis of citation sentiment improves the bibliometric measures. The previous method of evaluating an article's influence involved counting the citations. Nevertheless, citation sentiment analysis may be used to weigh each reference text by taking the feelings of the cited sources into account. The sentiments indicated in the text are typically obscured, making it challenging to determine whether they are negative, positive, or neutral (*Xu et al., 2015*). In terms of people, reading the citation text and identifying the sentiment represented in the citation text is simple. Yet, developing a model to predict the polarity of attitudes is a complex and time-consuming operation (*Lawani, 1982*). The majority of the reference text's emotion polarity appears to be neutral, with any concealed negative or positive feelings. Several methods may be utilised for hidden sentiment analysis, including analysis, feature-based, lexical and structure-based sentiment analysis, *etc*. The following major advances are made by this work, which uses machine learning to automatically classify in-text attitudes.

Pattern analysis of researchers' profiles regarding self-citation is an important research area these days. Many research papers and full-text availability in recent years have opened up new aspects for investigating the influence of citation analysis. A citation is a reference to another research publication that is included in a research article. Citing prior publications to substantiate assertions, dispute current assumptions, or build the foundations of a scientific notion is required in scientific writing (*Case & Higgins, 2000*). As a result, citations have become essential in evaluating the success of scientific work (*Umer et al., 2021*, *2022b*). Citations are commonly assumed to indicate the significance or brilliance of a research (*Aksnes, Langfeldt & Wouters, 2019*). What connection exists between the number of citations and the calibre of a research project? What is the foundation for these assumptions? Many years of scientometric study have been conducted

on these and related topics and new aspects have been investigated. Self-citations are key concepts in citation analysis since they are references or citations to a research publication written by the authors themselves. Many studies have been conducted to determine the underlying fundamental reasons or motivations for self-citations. Authors may reference their publications as a method to boost their exposure in their field of study, or they may self-cite frequently out of egotism (*Aksnes, 2003*). Self-citation can also be utilized to emphasize, update, or accurate findings from past publications (*Tagliacozzo, 1977*). It may also be used to boost a researcher's prestige in their field, albeit as *Lawani (1982)* points out, this is a subject that requires sociological analysis.

Citation counts are utilized to compute the impact factor of the journal and to evaluate academic production, which influences funding and career opportunities. They may also influence academics', organizations', and publications' reputations (*Foley & Della Sala, 2010*). Bibliometrics based on citation count may be subject to manipulation by employing tactics that make it appear untrustworthy (*Mavrogenis, Ruggieri & Papagelopoulos, 2010a*). Self-citation, in all of its expressions, may be considered such a tactic, and as a result, it has gained popularity in recent years. It is acceptable and even necessary to reference oneself.

Self-citation may be used as a marketing tool, enhancing the exposure of the researcher's work and generating subsequent cites from other sources. Since it might spread false notions, inaccurate information is deemed bad. Moreover, incorrect and overly frequent self-citations may skew academic literature and affect citation assessment metrics (*Mavrogenis, Ruggieri & Papagelopoulos, 2010a*). This is exacerbated by the fact that crucial criteria such as the i10 index and h-index are dependent on citation counts alone, and qualitative features are frequently overlooked. Self-citation literature has grown in popularity in recent years. The self-citation rate (SCR) may be calculated at the author, journal, and national levels. The SCRs for authors, journals, and nations ranged from 2.2% to 18% (*Lopez et al., 2016*), 6.35% to 11.85% (*Sundaram et al., 2020*), and 17.8% to 54.9% (*Shehatta & Al-Rubaish, 2019*) in many research publications that calculated self-citations. Between 1996 and 2008, national SCR grew in the majority of countries, most notably China, the United States, and *Jaffe (2011)*.

The following significant contributions are the subject of this study.

- Case 1 qualitatively explores citation sentiments and proposes an efficient feature representation method coupled with a supervised machine learning model to classify citation instances into positive, negative, or neutral categories.
- The effectiveness of the Synthetic Minority Oversampling Technique (SMOTE) in balancing the citation sentiment dataset is assessed.
- Case 2 considers direct and co-author citations to be self-citations and uses the top 2% data from 2020 to compute self-citations, with a focus on Pakistani scholars on the list. By calculating and evaluating the self-citation tendencies of the top-cited writers over the last five years.
- The study's findings, particularly the observation that citations from the Web of Science may overstate authors' self-citation ratios, provide a valuable contribution to the evaluation of existing citation data sources.

## RELATED WORK

Several criteria have been established over time to evaluate the calibre of a research publication or writers. For instance, the h-index is a crucial component in determining the author's relevance and position (*Hirsch, 2005*). In addition, impact factor and eigenvector are used for the same purpose *Pan & Fortunato (2014)* and *West et al. (2013)*. Both quantitative and qualitative methods of estimating an article's rank have not been thoroughly investigated. A very recent method of citation sentiment analysis may be used to analyse the significance of research publications without being constrained by the limits of quantitative methods. For example, *Kochhar & Ojha (2020)* measured the effect of a research publication using a hybrid model that combines objective and arbitrary criteria. The study linked citation feelings with the impact factors of the research article, the author, and both for this goal. In later phases, each lemma is tagged, and the SentiWordNet is used to determine its score.

Some research combines the sentiment of citations with objective measurements, whereas other approaches just consider the sentiment of citations. *Ikram & Afzal (2019)* identified the aspect-level sentiments and then suggested a two-level citation sentiment analysis technique. With the help of the material immediately surrounding the reference, several elements are initially derived from the citation phrases. A linguistic rule-based technique is utilised to determine the polarity of these factors in sentiment analysis. The support vector machine is used with N-gram features to achieve the highest level of sentiment categorization accuracy. Similarly, *Nguyen et al. (2017)* presented a deep learning system for sentiment analysis of the paper. LSTM approach is used with word embedding using word2vec while the data imbalance is dealt with using SMOTE. Findings indicate that the proposed system gives good results in contrast to the conventional SVM.

*Athar & Teufel (2012)* worked on context-based sentiment classification. To investigate how different approaches are affected by the context window's size, numerous experiments are run. The outcome with the N-gram features demonstrates that the addition of contexts expands the vocabulary and has an impact on performance. *Ghosh & Shah (2020)* investigate the importance of the features for ranking a scientific article. The study performed the citing sentiment on the ACL paper collection. Selected characteristics, including sentiment score, N-grams with negative and positive polarity, self-citation, a portion of search tags, and sentiment words are used to train the models and achieved the highest accuracy score of 80.61%.

The impact factor, a common statistic for analyzing scientific articles, has been the focus of several types of research throughout the years. Many researchers are attempting to determine the 'best' way for appraising research. Some researchers have focused on decreasing citation-based manipulations, while others have worked to establish fresh assessment approaches. In addition to editorial efforts, the issue of mandatory citations has lately been addressed. A wide variety of critical scientific choices, including employment, promotion, research funding allocations, and ranking, are becoming more reliant on citation count. According to *Aksnes (2003)*, the influence of self-citations is relatively minor. Author self-citations should be avoided below the micro level, according to the

authors. The research of *Glänzel & Thijs (2004)*, on the other hand, indicated that, while the influence of self-citations is very modest, they cannot be ruled out.

*Glänzel & Thijs (2004)* advocate for reporting citation impact statistics at the research institute level that include and omit author's self-citations. Instead of eliminating the author's self-citations at the micro and macro levels (individual researchers and research institutes), *Glänzel et al. (2006)* presented supplementary indications based on the author's self-citations. The authors demonstrated that while self-citations still provide valuable information, non-self-citations are the most significant citations for evaluation purposes (*Costas, van Leeuwen & Bordons, 2010*). *Hirsch (2005)* argues that self-citation should be avoided preferably at the micro level. Still, they also say that the author's self-citations have no effect on the h-index or have less impact than overall citation counts. *Hirsch (2005)* understates the h-index sensitivity to the author's self-citation in the h-index calculation, according to *Schreiber (2007)*, a stance held by *Vinkler (2007)* and *Gianoli & Molina-Montenegro (2009)*. *Simoes & Crespo (2020)* presented a system that employs self-citations as an informational resource, and they assess it using paper-based criteria, focusing on the new and larger idea of scientific impact.

According to *Fowler & Aksnes (2007)*, removing self-citation from the computation of the citation impact indicator may not be enough since self-citation may function as a promotion for researchers' work. They also conclude that each self-citation appears to be the outcome of 3.65 more citations from others. Previous research based on a smaller dataset by *Medoff (2006)* revealed no significant proof of an 'advertisement impact' of author self-citation. According to *Garfield (1997)*, for self-citation to be manipulative, it must be extravagant, and unimportant. However, given how much scientific disciplines rely on self-citation numbers, defining an 'appropriate' or even 'excessive' disciplinary rate is challenging (*Snyder & Bonzi, 1998*). In a study titled "How much is too much?" a group of academics asked this question. 'The distinction between research impact and excessive self-citation.' In the context of self-citations, the authors underline the significance of expert interpretation (*Szomszor, Pendlebury & Adams, 2020*).

While *Livas & Delli (2018)* identified no significant connection measures between self-citations and impact variables, cases of excessive self-citation are relatively rare (*Van Noorden & Chawla, 2019*). Clarivate now employs specialists to detect journals' very frequent self-citations, and those found to be guilty are removed from their Journal Citation Reports (JCR) (*WoS, 2023b*). Nonetheless, some circumstances still exist and have in the past. A total of 33 articles were suppressed by Clarivate in 2019 alone as a result of manipulating self-citations (*WoS, 2023a*). To overcome this issue, bold action is required, and some researchers advocate that the impact factor computation be self-citation free, or at the very least self-citation modified (*Mavrogenis, Ruggieri & Papagelopoulos, 2010b*).

Despite widespread criticism, *Zhao, Strotmann & Cappello (2018)* argue that self-citations serve a greater practical purpose than citations from other sources. Conversely, self-citations have a weaker correlation with academic influence and are less likely to be influential, according to *Zhu et al. (2015)*. In a recent research on the motivation for content-based citations, *Pride & Knoth (2017)* assigned a lesser value to self-citations, which was seen as a sign of insecurity or reluctance.

This study differs from others in that it seeks to ascertain if the impact factor is always the 'best' statistic to utilize as a measuring tool across all publications. To what degree are self-citations manipulative, and how much do they rely on a journal's size, publication location, topic matter, or language?

## CASE 1: ANALYSIS OF IN-TEXT SENTIMENT OF CITATIONS

### Materials and methods

The methodologies and approaches employed in this study are briefly covered in this section. Figure 1 shows the suggested approach's architectural layout. Starting with data retrieval, the approach follows the generation of fake text generation by utilizing the original dataset to make it balanced. Feature engineering is then carried out that involves term frequency (TF) and term frequency-inverse document frequency (TF-IDF). The data is divided into training and testing groups, and the chosen machine learning models are utilised to classify sentiment.

### Citation sentiment dataset

This study utilizes the 'citation sentiment *corpus*' dataset taken from ACL Anthology Network (*Athar, 2011*). The dataset contains 8,736 citation text annotated by human experts. The dataset comprises 'Source_Paper ID', 'Target_Paper ID', 'Citation_Text', and 'Sentiment'. The 'Source_Paper ID' is the citing paper's ID that represents the source of the text, 'Target_Paper ID' is the cited paper's ID, 'Citation_Text' is the original text that includes the citation while 'Sentiment' is the label of the target class and can be 'positive', 'negative', or 'neutral'. Many examples of the dataset's records are provided in Table 1.

### Machine learning classifiers

For the solution of the regression and classification problems, supervised machine learning algorithms are extensively used (*Safavian & Landgrebe, 1991*). Tree-based and regression-based algorithms are used in this study. This study used eight different supervised algorithms to solve the classification problem.

#### Decision tree

The decision tree (DT) is a supervised machine learning model that learns discrete rules from data features to predict target variables (*Sharma & Kumar, 2016*). The main benefit of the DT is the decision rules and features subset that appear at different classification steps. DT comprises different kinds of leaf nodes and various internal nodes having branches. Every leaf node denotes a target class while internal nodes denote features that are connected with branches to perform classification. The efficacy of the DT is based on how well-trained it is on the dataset.

#### AdaBoost classifier

AdaBoost from adaptive boosting is based on an ensemble learning classifier that utilizes the boosting method to train weak learners (*Zhang et al., 2014*). It combines many weak learners to recursively train them on the copies of the actual dataset, where every weak
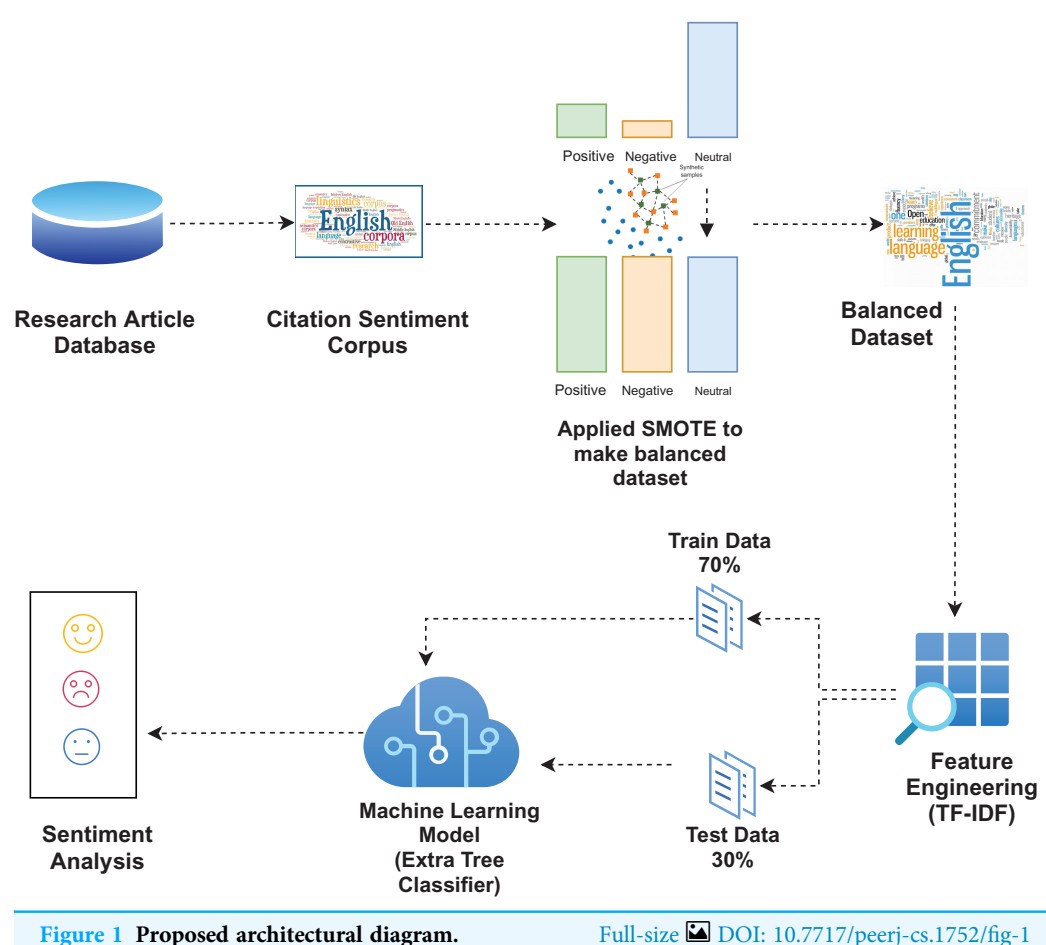

**Figure 1  Proposed architectural diagram.**  

**Table 1  Example of different sentiments from the citation sentiment *Corpus*.**

| Citation text | Sentiment |
| --- | --- |
| "One of the most effective taggers based on a pure HMM is that developed at Xerox (Cutting et al., 1992)." | Positive |
| "Jing & McKeown (2000) have proposed a rule-based algorithm for sentence combination, but no results have been reported." | Negative |
| "To contrast, Jing & McKeown (2000) concentrated on analyzing human-written summaries in order to determine how professionals construct summaries." | Neutral |

learner focuses on the outliers. It is a metadata model which takes the *N* number of weak learners and uses different assigned weights for training.

### Logistic regression

Logistic regression (LR) is a statistical algorithm that uses different variables to compute the final results. It is a regression-based model which estimates the class' probability. Therefore, it performs best for categorical data. LR uses a logistic function to estimate the probability and determine the relationship between dependant and independent variables (*Sebastiani, 2002*).

### Stochastic gradient classifier

The working of stochastic gradient classifier (SGD) is similar to the LR and SVM. For the multi-class classification, SGD proves to be a powerful classifier as it aggregates the various binary classifiers in the one-verses-all technique. SGD selects the examples from the batch randomly, so hyperparameters of SGD need correct values to achieve precise results. It is highly sensitive towards scaling of features (*Zadrozny & Elkan, 2002*).

### Random forest

Random forest (RF) comprises numerous decision trees which work separately to find the result while a majority of votes are used to decide the outcome. The outcome error rate is very less than other classifiers which is attributed to low low correlation among trees (*Gregorutti, Michel & Saint-Pierre, 2017*). Different split criteria are used for RF; the dataset is split based on the Gini index which is a cost function. The bagging technique is used in RF; in bagging several classifiers are trained using bootstrapped data, which are used to reduce the variance.

### Extra tree classifier

Extra tree classifier (ETC) uses the meta estimator, it trains a large number of weak learners of the random samples of the dataset which improves the result (*Rustam et al., 2019*). It is an ensemble model like RF widely utilized for classification problems. ETC differs from RF in the way of constructing of trees in the forest. It uses actual data for training, unlike RF which uses bootstrap data samples. At every node, a tree uses $k$ features of the random sample. Trees select the best feature for splitting. These random feature samples lead toward the multiple de-correlated DTs.

### Support vector classifier

Firstly proposed by Cortes and Vapnik, the support vector classifier (SVC) is a binary classification method that can be expandable to multi-class issues (*Cortes & Vapnik, 1995*). The SVC is used to deal with multi-class classification problems. To deal with nonlinear classification, outlier detection, and regression support vectors is a powerful technique. But the major drawback of SVC is that it relies on cross-validation of data, it performs poorly on small datasets.

### Voting classifier

Recently voting classifiers (VC) have shown better performance for many tasks than the traditional models. In a voting classifier, many classifiers can be added concerning training time constraints, and a single regression model is used as a regression model to determine the outcome of the vote. Each model predicts the target label and voting is performed between the classifiers to determine the target class label (*Catal & Nangir, 2017*). Soft and hard voting is used where soft voting considers the probability value of different classes from each classifier. In contrast, hard voting considers classifiers' prediction as votes and the winning class is the one with the most votes. This study combines LR and SGD as voting classifiers.

## Feature extraction

The technique of finding meaningful features from the data for good and efficient training of the machine learning model is known as feature engineering. Techniques for feature engineering can help machine learning algorithms perform better. Once significant features are extracted from the raw data using feature engineering, it helps to improve the learning algorithm's consistency and accuracy. This study used SMOTE upsampling, prediction-based (TF) (*He & Ounis, 2003*), and vectorization (TF-IDF) (*Christian, Agus & Suhartono, 2016*) features.

## Dealing with dataset imbalance

Data imbalance in datasets, where one class significantly outnumbers another, is a critical issue in data analysis, machine learning, and statistics. It leads to biased model performance, loss of valuable information about the minority class, reduced generalization to new data, ineffective evaluation metrics, and potentially high misclassification costs. Addressing data imbalance is crucial for accurate and valuable results in various real-world applications. Data imbalance can skew model predictions and obscure the significance of minority class issues. Techniques like SMOTE can mitigate these problems by balancing the dataset and enhancing the model's ability to generalize effectively. This study utilizes SMOTE to solve the problem of the imbalanced dataset.

### Using synthetic minority oversampling technique

SMOTE is a popular oversampling method for addressing the issue of unbalanced data. By generating the minority class's random syntactic data from its closest neighbours using Euclidian distance, SMOTE increases the number of instances (*Chawla, 2009*). Because they are created based on the original characteristics, newly generated instances are extremely comparable to the original data. To deal with high dimensional data SMOTE is not a good choice because it creates extra noise. A recent study uses the application of SMOTE for predicting people with heart failure (*Ishaq et al., 2021*). Machine learning and SMOTE show reasonable results but still do not quite well to compete with deep learning models (*Umer et al., 2022a*). This study uses SMOTE to generate a new training dataset.

## Results for case 1

Many classifiers are tested for performance using various assessment criteria for citation sentiment analysis. This study uses accuracy, precision, recall, and F1 score as the evaluation metrics. For the implementation of the machine learning algorithm, the sci-kit-learn library and NLTK have been utilized in Python. During training and testing, the data is divided into 0.7 to 0.3 ratios, respectively.

### Performance of classifiers using TF without SMOTE

The efficiency of the classifiers has been compared using TF without SMOTE for sentiment analysis of citation text. The voting classifier achieves the greatest accuracy of 0.9122, according to the results shown in Table 2. SVC had a 0.8961 accuracy score, which was the second-highest. In terms of accuracy, recall, and F1 score for citation sentiment analysis,

**Table 2 Classification result of classifiers models using TF without SMOTE.**

| Models | Acc. | Prec. | Recall | F1 |
|---|---|---|---|---|
| DT | 0.8473 | 0.84 | 0.85 | 0.84 |
| AdaBoost | 0.8752 | 0.85 | 0.88 | 0.85 |
| LR | 0.8714 | 0.84 | 0.87 | 0.82 |
| SGD | 0.8870 | 0.87 | 0.89 | 0.86 |
| RF | 0.8760 | 0.84 | 0.88 | 0.84 |
| ETC | 0.8775 | 0.85 | 0.88 | 0.84 |
| SVC | 0.8961 | 0.87 | 0.89 | 0.87 |
| VC | 0.9122 | 0.90 | 0.90 | 0.90 |

LR, RF, and ETC produce findings that are quite comparable. Nonetheless, DT exhibits the lowest outcomes among all TF models, with an accuracy rating of 0.8473.

### Performance of classifiers using TF with SMOTE

Supervised machine learning models have been evaluated using TF features with SMOTE. It is evident from Table 3 that all classifiers performed much better for sentiment analysis of citation text when TF and SMOTE were combined. The best model continues to be RF, which earns accuracy scores of 0.9729, precision scores of 0.98, recall scores of 0.96, and F1 scores of 0.97. For each evaluation parameter, an accuracy greater than 0.90 is displayed by DT, LR, SGD, RF, and SVC. With accuracy, precision, recall, and F1 score values of 0.8361, 0.84, 0.79, and 0.82 respectively, AdaBoost performs the lowest.

### Performance of classifiers using TF-IDF without SMOTE

Without utilising SMOTE, the outcomes of classifiers that use the feature extraction method TF-IDF are compared. The accuracy, precision, recall, and F1 score comparison of classifiers employing TF-IDF is shown in Table 4. With an accuracy score of 0.9122 and 0.90 values for precision, recall, and F1, it can be shown that the voting classifier performs better than other models. With a 0.8961 accuracy score, 0.87 precision, 0.89 recall, and 0.87 F1 score, SVC performs just significantly worse. For citation sentiment analysis, RF, and ETC produce comparable findings with accuracy scores of 0.8760 and 0.8775, respectively.

### Performance of classifiers using TF-IDF with SMOTE

After using SMOTE, the performance of the models is also assessed using the TF-IDF. The comparison of classifiers utilising TF-IDF with SMOTE balanced dataset to assess sentiments of citation text is shown in the results presented in Table 5. Classifiers that use TF-IDF with SMOTE perform better than classifiers that use TF-IDF without SMOTE. RF achieved the best results with a 0.9729 accuracy score, 0.98 precision, 0.96 recall, and 0.97 F1 score. All models have shown significant improvement in classification accuracy after applying SMOTE. SVC achieved values higher than 0.96 in terms of all evaluation measures.

**Table 3 Classification result classifiers using TF with SMOTE.**

| Models | Acc. | Prec. | Recall | F1 |
|--------|------|-------|--------|-----|
| DT | 0.9010 | 0.90 | 0.90 | 0.90 |
| AdaBoost | 0.8361 | 0.84 | 0.79 | 0.82 |
| LR | 0.9388 | 0.94 | 0.94 | 0.94 |
| SGD | 0.9361 | 0.96 | 0.96 | 0.96 |
| RF | 0.9729 | 0.98 | 0.96 | 0.97 |
| ETC | 0.8444 | 0.84 | 0.84 | 0.84 |
| SVC | 0.9669 | 0.97 | 0.97 | 0.97 |
| VC | 0.8667 | 0.86 | 0.87 | 0.86 |

**Table 4 Classification result of classifiers using TF-IDF without SMOTE.**

| Models | Acc. | Prec. | Recall | F1 |
|--------|------|-------|--------|-----|
| DT | 0.8473 | 0.84 | 0.85 | 0.84 |
| AdaBoost | 0.8752 | 0.85 | 0.88 | 0.85 |
| LR | 0.8714 | 0.84 | 0.87 | 0.82 |
| SGD | 0.8870 | 0.87 | 0.89 | 0.86 |
| RF | 0.8760 | 0.84 | 0.88 | 0.84 |
| ETC | 0.8775 | 0.85 | 0.88 | 0.84 |
| SVC | 0.8961 | 0.87 | 0.89 | 0.87 |
| VC | 0.9122 | 0.90 | 0.90 | 0.90 |

**Table 5 Classification result classifiers using TF-IDF with SMOTE.**

| Models | Acc. | Prec. | Recall | F1 |
|--------|------|-------|--------|-----|
| DT | 0.9010 | 0.90 | 0.90 | 0.90 |
| AdaBoost | 0.8361 | 0.84 | 0.79 | 0.82 |
| LR | 0.9388 | 0.94 | 0.94 | 0.94 |
| SGD | 0.9361 | 0.96 | 0.96 | 0.96 |
| RF | 0.9729 | 0.98 | 0.96 | 0.97 |
| ETC | 0.8444 | 0.84 | 0.84 | 0.84 |
| SVC | 0.9669 | 0.97 | 0.97 | 0.97 |
| VC | 0.8667 | 0.86 | 0.87 | 0.86 |

## Comparative analysis with cutting-edge methods

The proposed model's performance is compared with state-of-the-art research work based on feature engineering and learning models (*Karim et al., 2022*). The reason for selecting this research for comparison is that this research work also utilized four different types of feature engineering for optimizing the performance of citation sentiment analysis. Table 6 displays the results of the models, which reveal that the CNN model using combined

**Table 6 Comparative analysis of the proposed approach.**

| Model | Accuracy | Precision | Recall | F1 score |
|---|---|---|---|---|
| Classification results of classifiers using fastText (*Karim et al., 2022*) | | | | |
| CNN | 0.89 | 0.87 | 0.85 | 0.86 |
| LSTM | 0.86 | 0.70 | 0.74 | 0.72 |
| RF | 0.86 | 0.79 | 0.86 | 0.81 |
| SGD | 0.86 | 0.76 | 0.87 | 0.81 |
| LR | 0.86 | 0.76 | 0.87 | 0.81 |
| Classification results of classifiers using fastText subword (*Karim et al., 2022*) | | | | |
| CNN | 0.87 | 0.85 | 0.82 | 0.83 |
| LSTM | 0.87 | 0.84 | 0.80 | 0.81 |
| RF | 0.87 | 0.82 | 0.87 | 0.83 |
| SGD | 0.87 | 0.76 | 0.87 | 0.82 |
| LR | 0.87 | 0.76 | 0.87 | 0.81 |
| Classification results of classifiers using GLOVE (*Karim et al., 2022*) | | | | |
| CNN | 0.86 | 0.88 | 0.89 | 0.88 |
| LSTM | 0.84 | 0.78 | 0.74 | 0.76 |
| RF | 0.85 | 0.81 | 0.86 | 0.80 |
| SGD | 0.85 | 0.73 | 0.86 | 0.79 |
| LR | 0.85 | 0.76 | 0.85 | 0.79 |
| Classification results of using combined features (*Karim et al., 2022*) | | | | |
| CNN | 0.93 | 0.94 | 0.96 | 0.95 |
| LSTM | 0.89 | 0.91 | 0.93 | 0.92 |
| RF | 0.87 | 0.89 | 0.92 | 0.90 |
| SGD | 0.88 | 0.91 | 0.91 | 0.91 |
| LR | 0.91 | 0.90 | 0.88 | 0.89 |
| Classification results of proposed model | | | | |
| **Proposed model** | **0.97** | **0.98** | **0.96** | **0.97** |

features yields the highest performance than other models with 0.93 value of accuracy, 0.94 value of precision, 0.96 value of recall, and 0.95 value of F1 score. On the other hand, the proposed model (RF using TF-IDF with SMOTE) has shown superior performance with 0.9729 value of accuracy, 0.98 value of precision, 0.96 value of recall and 0.97 value of F1 score.

## Limitations of the proposed model

The proposed model, RF using TF-IDF with SMOTE, exhibits strong performance in sentiment analysis of citation text, as indicated by high accuracy, precision, recall, and F1 score. However, potential limitations include concerns about overfitting due to the small dataset, sensitivity to hyperparameters, and lack of information on feature importance. The model's generalizability to external datasets and domains remains unexplored.

## CASE 2: ANALYZING SELF-CITATION PATTERNS AMONG TOP 25 PAKISTANI RESEARCHERS

This study examines self-citations for the top 25 Pakistani researchers. Figure 2 depicts the process diagram of the adopted technique. It begins with data from the top 2% of researchers on the Web of Science (WoS), followed by data extraction for the top 25 Pakistani researchers is referred in the Table 7. The writers are ordered based on their self-citation ratio. Following that, a fresh dataset is gathered for the selected authors' further investigation. The suggested technique and data are used to estimate direct and indirect self-citation. Finally, the authors' modified impact factor is computed.

### Data selection

To begin, the top 2% researcher statistics supplied by the Stanford-Scopus/Elsevier cooperation for 2020 are used (*Ioannidis, Boyack & Baas, 2020*). The database includes standardised data on citations, h-indexes, co-authorship adjusted hm-index, citations to works with various authors, and a composite indicator for more than 100,000 notable scientists. There are 22 scientific domains and 176 sub-fields for scientists. For all scientists who have published at least five publications, field and subfield-specific percentiles are also supplied. The top 100,000 researchers are chosen based on their c-score (with and without self-citations) or a percentile rank of 2% or above.

### Data collection

Several bibliometric measures are affected by self-citation. Due to information scarcity, analyzing relevant data on self-citation impacts on global ranking and their influences on SCR is difficult. To accomplish this purpose, this study selects the top 25 Pakistani researchers among the top 2% of researchers in 2020. The study's major focus is on researchers that have a self-citation ratio of 50% or above. Mentioning an author's previous work in his present work is known as self-citation. However, the WoS eliminates only citations made by the manuscript's first author. However, citations from any author of the present article are treated as self-citations in this study. To evaluate self-citation at this level, information is needed from all the publications that mentioned paper A, for example, as well as the author list to discover the self-citation. This study gathered paper information from the top 25 Pakistani researchers in 2020. It is impossible to collect information on all such articles because numerous writers have written more than 100 papers apiece. Only the most recent ten articles published in 2020 are considered for this purpose. This study explored the influence of the primary author's self-citation on his profile and the effect of his co-author's citations.

### Proposed approach

This research considers a citation to be self-citation if the article's primary author or a secondary author (any author in the author list other than the lead author) references a paper in any of their previous papers. Figure 3 shows an example instance for greater explanation. The document considered for self-citation is referred to as a "source paper," and the publications that reference the "source paper" are referred to as "citing papers". It

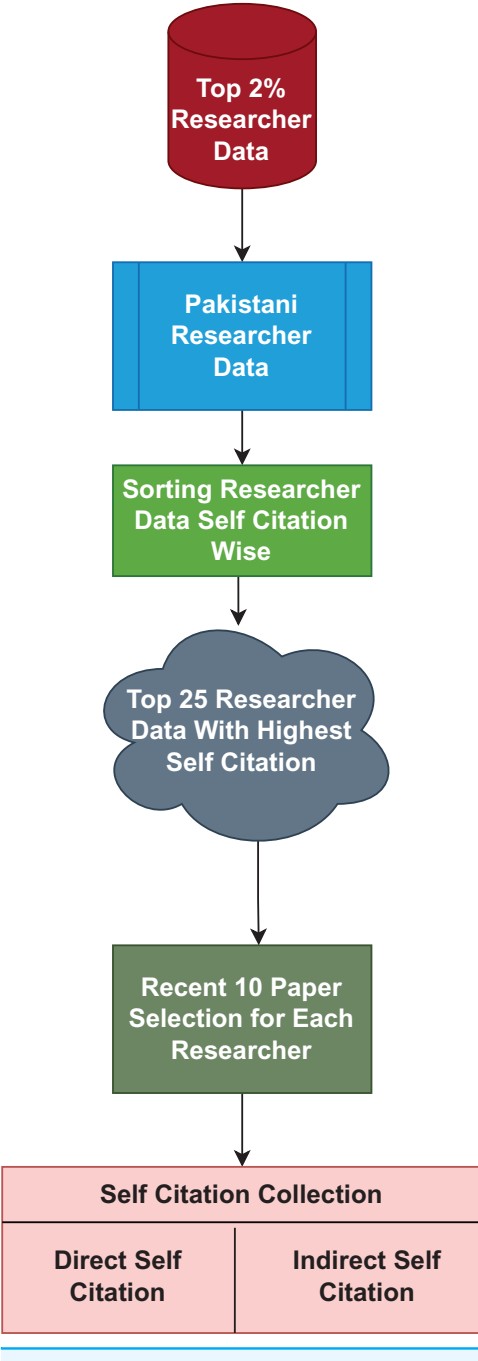

**Figure 2** Steps to perform self-citation analysis.

is referred to as direct self-citation when the primary author of a work mentions their paper. It is known as indirect self-citation when any author on a paper's author list cites that paper. self-citations are classified as direct or indirect in this study. Because such statistics are not accessible, this study compiles a dataset of the top 25 Pakistani researchers from the top 2% researchers' ranking.

**Table 7 Self-citation dataset for top 25 Pakistani researchers.**

| Source paper | Citing paper 1 | Citing paper 2 | Citing paper 3 | ,…, | Citing paper N |
|---|---|---|---|---|---|
| Z. Yousaf, M. Z. Bhatti, T. Naseer | M. Sharif, T. Naseer | M. Sharif, T. Naseer | M. Sharif, T. Naseer | ,…, | M. Sharif, T. Naseer |
| A. Zada, N. Ali, M. Ateeq, A. M. Huerta-Flores, Z. Hussain, S. Shaheen | K. Qi, S. Liu, A. Zada | A. Zada, M. Khan, M. A. Khan, Q. Khan | H. Yasmeen, A. Zada, S. Ali, I. Khan, W. Ali | ,…, | S. Ahmed, T. Arshad, A. Zada, A. Afzal, M. Khan |
| F. Hussain, G. Shabbir, S. Malik, M. Ramzan, A. H. Kara | M. Ali, F. Hussain, G. Shabbir, S. F. Hussain | S. Malik, F. Hussain, G. Shabbir | F. Hussain, G. Shabbir, S. Malik | ,…, | S. Malik, F. Hussain, G. Shabbir |

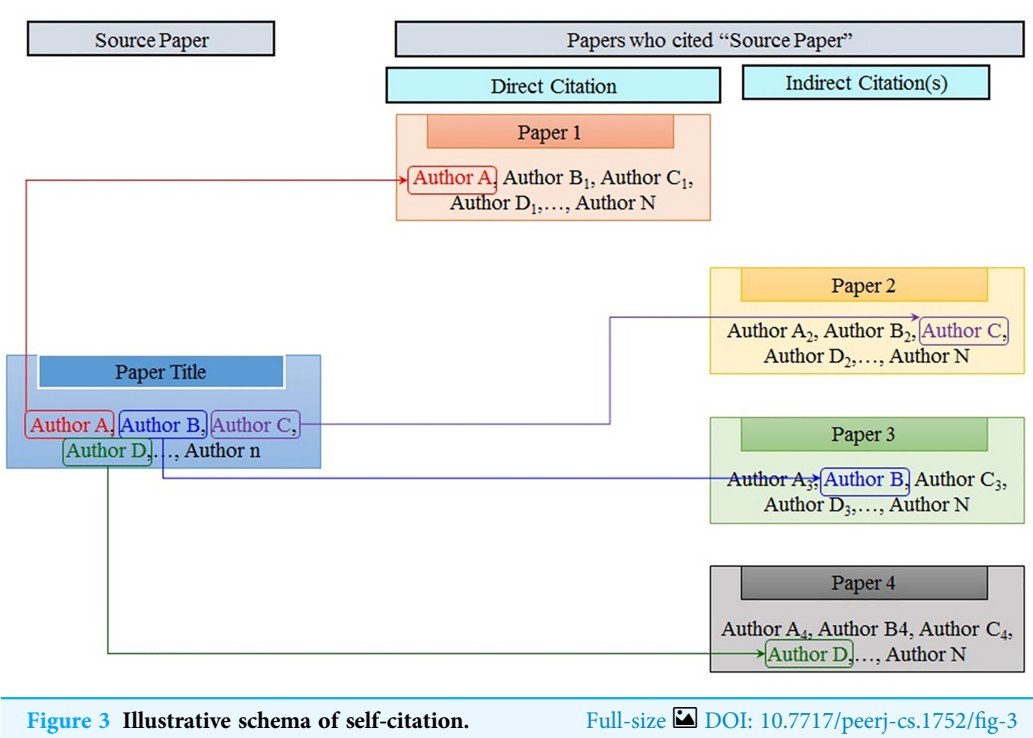

**Figure 3 Illustrative schema of self-citation.**

The data collected for the top 25 Pakistani researchers is made public, including author names, publications published in 2020, author names for citing works, and so on. Researchers' statistics are kept up to current until the end of 2020. The dataset and code may be found at the link (https://github.com/MUmerSabir/SelfCitation).

This study considers an article to be self-cited not just by the article's original author, but also by any of the co-authors who cite the piece. Once the data is obtained, Algorithm 1 is designed to collect self-citations from the dataset for all publications written by a certain researcher.

## Self-citation extraction

Self-citation extraction is carried out using Algorithm 1. The collected dataset is fed into the algorithm and it provides the list of self-cations for each author. Direct self-citations and indirect self-citations are calculated separately to analyze the change in the author's

| Algorithm 1 | Algorithm to find self-citation count. |
|---|---|

**Input:** Collected dataset for top 25 researchers

**Output:** Self citation count

1:  **for** $A = 1$ to $n$ **do**
2:      **for** $SP_{A_i} = 1$ to $P$ **do**
3:          $AL_{SP} \leftarrow getAuthorList(SP_{Ai})$
4:          **for** $CP_{SP_j} = 1$ to $Q$ **do**
5:              $AL_{CP} \leftarrow getAuthorList(CP_{SPi})$
6:              **if** $AL_{SP}.index[1] = AL_{CP}.index[AuthorName]$ **then**
7:                  $F_{ObjectNode} = key$
8:              **else if** $AL_{SP}.index[AuthorName] = AL_{CP}.index[AuthorName]$
9:                  $C_{ObjectNode} = key$
10:             **end if**
11:         **end for**
12:     **end for**
13: **end for**

citation profile. The dataset contains the author list for the source paper against the author's lists for all the citing papers, where $n$ is the number of authors considered for analysis which is 25 in this study. $SP_{A_i}$ indicates the source paper $i$ for an author while $CP_{SP_j}$ shows the citing paper for source paper $j$ of an author. Each author's papers are traversed to calculate self-citations (Lines 1 to 12). Line 3 gets the author list of paper $j$ of author $A_i$. Lines 4 to 10 obtain the author's list of all citing papers and compare the author list of the source paper. For direct self-citation, only the first author is searched in the citing paper authors list (Line 6). Line 8 is used to obtain indirect self-citation. Separate records are maintained for direct and indirect self-citations.

## Findings for case 2

This study examines the influence of self-citation from a variety of angles. The first part of this section discusses the results for the nation, institution, and journal utilizing data from the top 2% of researchers in several characteristics. Self-citation data, for example, are studied in terms of domain, sub-domain, nation, institution, and so on. Similarly, self-citation patterns are explored concerning journal quartile ranking. The second section displays the findings from the data collected from the top 25 Pakistani researchers.

### Impact of self citations on journals

This study analyzes the impact of self-citations on the quartile of SCIE-indexed journals. For this purpose, the top 25 journals with the highest number of publications are considered in each quartile. Figure 4 shows the impact factor of quartile-1 (Q1) journals considering self-citations. It can be seen that the ratio of self-citation is minor as the impact
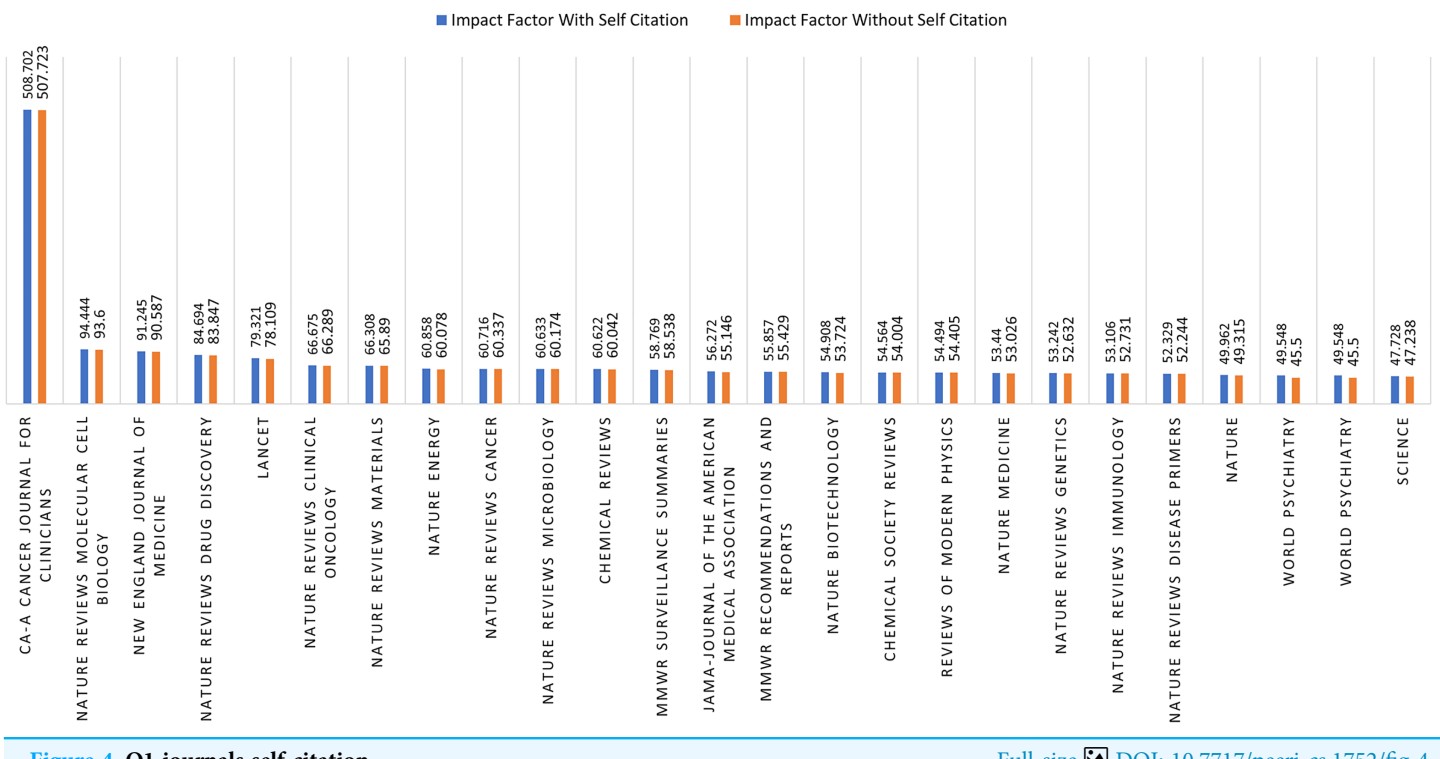

**Figure 4  Q1 journals self-citation.**

factor of Q1 journals is almost the same with and without self-citation. However, the same is not true for quartile-2 (Q2) journals, as a clear difference, can be observed in the impact factor of journals if self-citations are excluded, as shown in Fig. 5. For example, the impact factor of the Nanotechnology Reviews journal reduces from 7.848 to 3.828 if self-citations are excluded. The majority of the journals are found to have a substantial change in the impact factor for Q2 journals.

Quartile-3 (Q3) journals are observed to exhibit the same phenomenon when self-citation is considered. A few journals are found to have a significant change in the impact factor when self-citations are not included like 'HLA', 'Journal of Cellular Biochemistry', and 'Journal of Sustainable Cement-based Materials', *etc.* while others have a marginal change in impact factor like 'Nanoscale Research-Letters', 'Nanoscale Advances', *etc.*

### Results using self-collected data

This section explains the findings achieved by applying Algorithm 1 on data collected from the top 25 Pakistani researchers. Figure 6 depicts the total number of citations received by the top 25 scholars in 2020. The data included in the graph comes from the top 2% of researchers. The chart only displays the researchers' total citations, eliminating self-citations. Choudhary, M. Iqbal of the University of Karachi received the most citations in 2020.
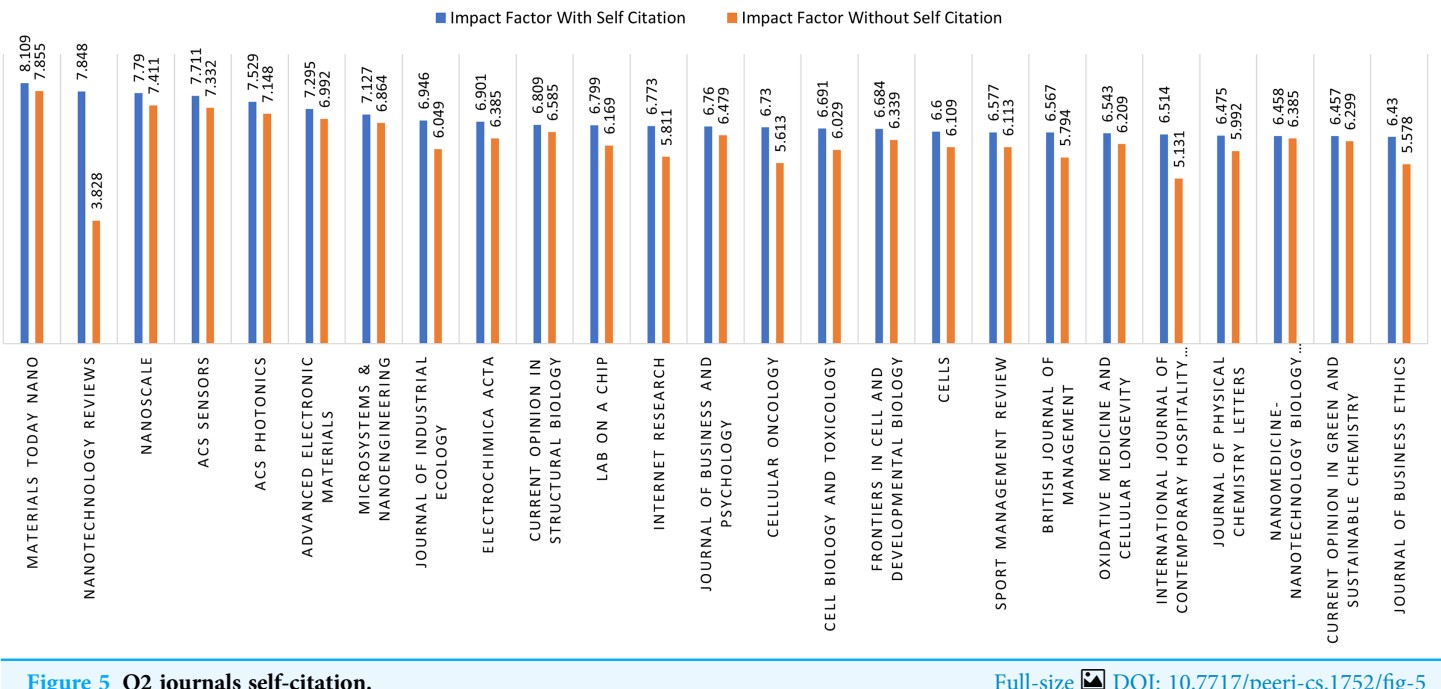

**Figure 5** Q2 journals self-citation.

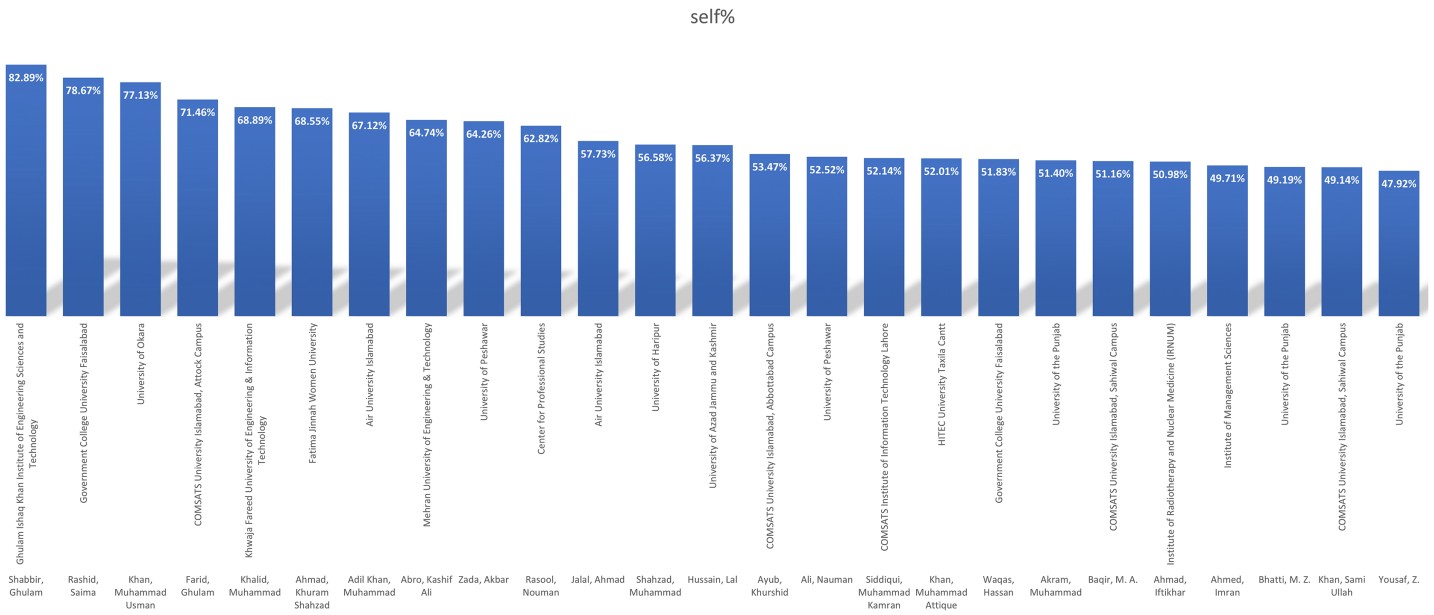

**Figure 6** Pakistani researcher's top 25 total citations.

Figure 7 depicts the top 25 writers in terms of self-citation. The authors are chosen based on the self-citation ratio found in the datasets of the top 2% of researchers. The data is collected for these researchers and analyzed further.
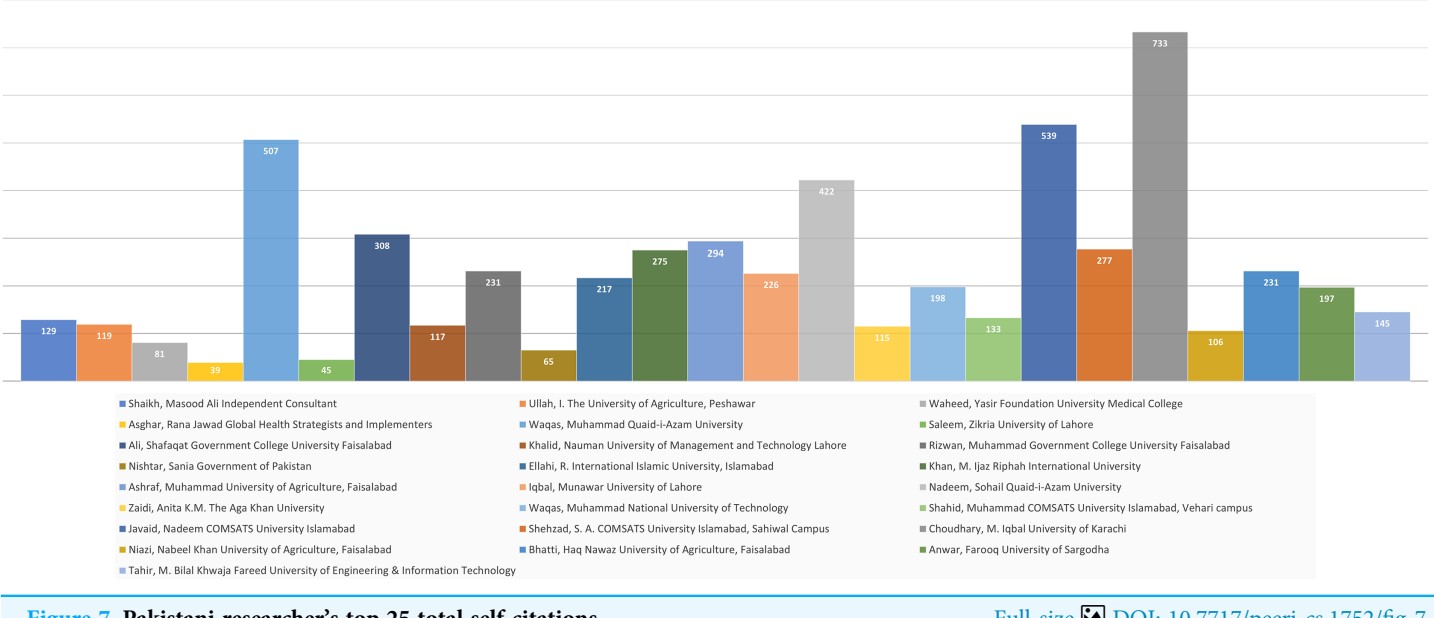

**Figure 7  Pakistani researcher's top 25 total self-citations.**

Table 8 displays the computed direct and indirect self-citations using the gathered data and Algorithm 1. The ratio of citations determined using the criteria stated in Algorithm 1 is shown in the columns direct and indirect self-citations. These columns differentiate the ratios of direct and indirect self-citations. The total self-citation column, on the other hand, displays the aggregate self-citation ratio of researchers and includes both direct and indirect self-citation. The results demonstrate that the total self-citation ratio for the selected researchers is exceptionally high, reaching more than 70% in certain cases. For the selected 25 researchers, the mean value for overall self-citation is 54.4096.

The maximum rate of direct self-citation was found to be 76.66%, while the lowest percentage was determined to be 6.66%. For the selected 25 researchers, the mean value for direct self-citation is 34.7856. The rate of indirect self-citation is larger than the rate of direct self-citation. The mean number for indirect self-citation percentage, for example, is 49.1464%, with the top and lowest percentages being 97.72% and 13.30%, respectively.

### Self citation comparison with top 2% researchers data

The complete data of three of the top 25 Pakistani researchers is collected to compare the difference between the self-citation count supplied by the top 2% researchers' data and the self-citation count gained in this study is referred in the Table 9. The gathered data includes all publications by selected authors for the year 2020, together with a complete author list and author lists for all papers that mentioned it. This data me be used to determine each author's direct and indirect self-citations and compare them to self-citations collected from WoS for the year 2020. The proportion of self-citations reported by the WoS is substantially lower than the one computed in this study. WoS only evaluates the self-citation of an article's initial author, which is unrealistic because other authors of a

**Table 8 Direct and indirect self-citation for top 25 Pakistani authors.**

| Author name | Institute name | Direct self citation % | Indirect self citation % | Overall self citation % |
|---|---|---|---|---|
| Ghulam Shabbir | Ghulam Ishaq Khan Institute of Engineering Sciences and Technology | 76.13% | 97.72% | 97.72% |
| Saima Rashid | Government College University Faisalabad | 40.94% | 62.99% | 72.44% |
| Muhammad Usman Khan | University of Okara | 22.40% | 68.80% | 74.40% |
| Ghulam Farid | COMSATS University Islamabad, Attock Campus | 57.44% | 64.89% | 67.02% |
| Muhammad Khalid | Khwaja Fareed University of Engineering & Information Technology | 11.11% | 50.00% | 51.58% |
| Khuram Shahzad Ahmad | Fatima Jinnah Women University | 40.86% | 44.08% | 44.08% |
| Muhammad Adil Khan | Air University Islamabad | 6.66% | 13.30% | 23.33% |
| Kashif Ali Abro | Mehran University of Engineering & Technology | 71.12% | 71.83% | 74.64% |
| Akbar Zada | University of Peshawar | 48.0% | 55.20% | 61.60% |
| Nouman Rasool | Center for Professional Studies | 16.50% | 46.6% | 51.45% |
| Ahmad Jalal | Air University Islamabad | 76.66% | 76.66% | 77.46% |
| Muhammad Shahzad | University of Haripur | 30.70% | 48.03% | 48.81% |
| Lal Hussain | University of Azad Jammu and Kashmir | 14.92% | 22.38% | 25.37% |
| Khurshid Ayub | COMSATS University Islamabad, Abbottabad Campus | 30.70% | 44.73% | 47.36% |
| Nauman Ali | University of Peshawar | 42.37% | 61.86% | 66.10% |
| Muhammad Kamran Siddiqui | COMSATS Institute of Information Technology Lahore | 17.59% | 29.62% | 53.70% |
| Muhammad Attique Khan | HITEC University Taxila Cantt | 49.24% | 59.84% | 62.87% |
| Hassan Waqas | Government College University Faisalabad | 20.30% | 31.57% | 48.87% |
| Muhammad Akram | University of the Punjab | 43.38% | 47.79% | 55.88% |
| M. A. Baqir | COMSATS University Islamabad, Sahiwal Campus | 38.09% | 52.38% | 54.76% |
| Iftikhar Ahmad | Institute of Radiotherapy and Nuclear Medicine (IRNUM) | 14.66% | 38.66% | 46.66% |
| Imran Ahmed | Institute of Management Sciences | 23.65% | 30.10% | 31.18% |
| M. Z. Bhatti | University of the Punjab | 50.35% | 69.5% | 75.17% |
| Sami Ullah Khan | COMSATS University Islamabad, Sahiwal Campus | 25.87% | 40.58% | 47.79% |

paper also mention the work and must be included in the self-citation, as done in this study. As a result, the proportion of self-citation calculated in this study is larger than that calculated in WoS. However, for the sake of self-citation estimate, it is more transparent and genuine. Table 10 presents the description of terms used in the paper.

### Comparative analysis of self-citation patterns

The approach used in Case 2 is compared with a study (Budimir et al., 2021) examining self-citation patterns. This study examines self-citation patterns for Slovenian scientific documents in six research fields over a more extended period (1996 to 2020) and includes more than 12,000 registered researchers in Slovenia. It compares self-citation patterns in Scopus and WoS databases for Slovenian scientific documents, and it evaluates the effects

**Table 9 Self-citation comparison for selected authors.**

| Author | #Of papers | Total citations | Direct self citations | Overall self citations | Self citation % | WoS self citation % |
|---|---|---|---|---|---|---|
| Saima Rashid | 53 | 1,457 | 471 | 738 | 50.65 | 16.20 |
| M. Usman Khan | 35 | 1,160 | 191 | 685 | 59.05 | 6.01 |
| Ghulam Shabbir | 11 | 102 | 71 | 100 | 98.03 | 78.16 |

**Table 10 Acronym table**

| Acronym | Description |
|---|---|
| CNN | Convolutional neural network |
| DT | Decision tree |
| ETC | Extra tree classifier |
| JCR | Journal citation reports |
| LR | Logistic regression |
| RF | Random forest |
| SCIE | Science citation index expanded |
| SCR | Self-citation rate |
| SGD | Stochastic gradient decent |
| SMOTE | Synthetic minority oversampling technique |
| SVC | Support vector classifier |
| TF-IDF | Term frequency-inverse document frequency |
| VC | Voting classifier |
| WoS | Web of science |

of self-citations on citations. It reports an average self-citation rate of about 22% for Slovenian researchers. It provides insights into the percentage of co-authorship with researchers from other countries.

While, the proposed study provides detailed data on self-citation patterns, including direct and indirect self-citations, making it valuable for researchers interested in understanding the nuances of self-citation behavior. It has a very specific focus on self-citation patterns among the top 25 Pakistani researchers in a particular year (2020). This narrow focus allows for a detailed and in-depth analysis of self-citation behaviors within this specific group. The self-citation ratios exceeding 70% in certain cases and a mean value of 54.4096% highlight the significance of self-citation practices among these top Pakistani researchers. It also mentions the use of Algorithm 1, which may provide a standardized and reproducible methodology for analyzing self-citation data. Furthermore, it focuses on a specific year (2020) and provides recent insights into self-citation behavior within the academic community, which may be of particular relevance to scholars and institutions.

## CONCLUSION

This research investigates the intricate world of citation sentiment analysis, shedding light on the profound impact of self-citation on an author's scientific profile. Citation count is

important to determine several metrics related to both author and journal performance. For research profile analysis, self-citations are excluded from total citations to provide a more transparent profile. Case 1 qualitatively analyzes citation sentiments, highlighting the importance of understanding the appreciation, critique, and foundational aspects embedded within scholarly references. It emphasizes the sentiment analysis of in-text citations by employing a machine learning model with an appropriate feature engineering technique. Results prove that random forest with TF-IDF on the SMOTE balanced dataset achieved a 0.9729 accuracy score.

In Case 2, the quantitative investigation takes centre stage, revealing direct and indirect self-citation patterns. Although self-citation is a common phenomenon, excessive self-citation leads to elevated h-index and exaggerated researcher profiles. This study proposes a model for calculating the self-citation of the top 25% Pakistani researchers in the world's top 2% researchers' data of 2020 and considers both direct and indirect self-citations. Results indicate that the self-citation count from the WoS is significantly different from the self-citations obtained using the proposed technique. It is so because the WoS considers the first author's citation as a self-citation. In future, authors intend to work on a modified h-index using the self-citation data gathered in this study.

### Funding
The funding is supported by Turki Aljrees with the support of the University of Hafr-Al Batin. The funders had no role in study design, data collection and analysis, decision to publish, or preparation of the manuscript.

### Grant Disclosures
The following grant information was disclosed by the authors:
University of Hafr-Al Batin.

### Competing Interests
Ali Kashif Bashir is an Academic Editor for PeerJ.

### Author Contributions
- Muhammad Umer conceived and designed the experiments, performed the experiments, analyzed the data, performed the computation work, prepared figures and/or tables, authored or reviewed drafts of the article, and approved the final draft.
- Turki Aljrees conceived and designed the experiments, performed the experiments, prepared figures and/or tables, authored or reviewed drafts of the article, and approved the final draft.
- Saleem Ullah conceived and designed the experiments, performed the experiments, analyzed the data, performed the computation work, prepared figures and/or tables, and approved the final draft.

- Ali Kashif Bashir conceived and designed the experiments, performed the experiments, prepared figures and/or tables, authored or reviewed drafts of the article, and approved the final draft.

## Data Availability

The data is available at GitHub and Zenodo: https://github.com/MUmerSabir/SelfCitation.

MUmerSabir. (2023). MUmerSabir/SelfCitation: DOI Request (as). Zenodo. https://doi.org/10.5281/zenodo.8267119.

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
