# Peer review of "Novel approach for quantitative and qualitative authors research profiling using feature fusion and tree-based learning approach"

_PeerJ Computer Science, doi:10.7717/peerj-cs.1752_

## Round 0.1 · original submission · Major Revisions

The authors should revise and provide detailed response to reviewer comments.

**Language Note:** The review process has identified that the English language must be improved. PeerJ can provide language editing services - please contact us at [email protected] for pricing (be sure to provide your manuscript number and title). Alternatively, you should make your own arrangements to improve the language quality and provide details in your response letter. – PeerJ Staff

·

Basic reporting

The theme of the paper is pretty much interesting and they way authors provided a comprehensive analysis of citation analysis is appreciate able. Still, the paper has some short coming that needed to be resolved before making any final decision on it.

1. There is no sub-heading and typo mistake on line number 198.
2. In figure 1, citation corpus and balance dataset symbol is not appropriate.
3. line 254-255 needed some more explanation about data imbalance why it is important and what it causes.
4. Add limitations of the proposed model of case 1

5. Case 2 line 310, heading is not self explainable.

6. Discourage the use of words like ‘ours’ in entire paper.
7. line 366, heading is not appropriate. It should be findings rather than results.

8. Table 8 needed to be restructured according to format.
9. Comparison needed to be extensive explaining both phases of the research paper.

10. English language editing’s are most important to get published in SCI-E indexed journals.

Overall paper is interesting and has potential but requires some major revisions.

Experimental design

Check comments on basic reporting

Validity of the findings

Check comments for basic reporting

Additional comments

Comments mentioned in basic reporting

·

Basic reporting

I have read the entire paper in a single sitting and for me the paper is very interesting. The best thing about this paper is that it covers both aspects of citation that majorly influence any researcher profile. The paper is written in well-mannered way and has the ability to be published in prestigious journal like PEERJ CS. My only suggestion about this paper is to add some related work on the case 2 too. Major portion of related work covers case 1 but no latest work on case 2 is presented. That is the only lack I found. Rest paper is okay for me.

Experimental design

work is perfect

Validity of the findings

perfect

Additional comments

perfect

Reviewer 3 ·

Basic reporting

The paper written by umer et al. is interesting in terms of novelty and contribute to the scientific research community. But, this research requires some major changes to further improve its quality.
1. In abstract, give equal importance to both cases by adding more details of case 1.
2. Major contributions points needed improvements to better highlight the cases of self citation analysis. Make 4~5 points rather than two.
3. Remove mistakes like on line number 112 akram and akram, line 121 ather ather, line 124 ghosh ghosh. Remove all mistakes like this in entire paper.
4. Feature engineering techniques TF, TF-IDF is mentioned in the line 180-186 but no reference or details are given for them.
5. Line 198, decision tree heading is missing.
6. Limit each machine learning model with 1 reference only.
7. Acronym table is necessary.
8. Try to add abbreviation of each term before its first usage.
9. Re-write conclusion and mention both cases equally in that.
10. English editing’s and proofing is necessary

Experimental design

Check basic comments

Validity of the findings

Check basic comments

Additional comments

Check basic comments

---

## Round 0.2 · accepted · Accept

The authors have revised the article as per reviewer comments.

·

Basic reporting

I believe it has been reviewed and edited in a good manner

Experimental design

Design is clear and fulfills the requirement

Validity of the findings

Great

Reviewer 3 ·

Basic reporting

All of my comments are resolved and no further comments from my side!

Experimental design

N/a

Validity of the findings

N/a

Additional comments

N/a